# Connective Tissue Disorders and Fragile X Molecular Status in Females: A Case Series and Review

**DOI:** 10.3390/ijms23169090

**Published:** 2022-08-13

**Authors:** Merlin G. Butler, Waheeda A. Hossain, Jacob Steinle, Harry Gao, Eleina Cox, Yuxin Niu, May Quach, Olivia J. Veatch

**Affiliations:** 1Department of Psychiatry & Behavioral Sciences, University of Kansas Medical Center, 3901 Rainbow Blvd. MS 4015, Kansas City, KS 66160, USA; 2Fulgent Genetics, 4978 Santa Anita Ave., Temple City, CA 91780, USA

**Keywords:** *FMR1* gene, fragile X syndrome, *FMR1* gray zone or intermediate alleles, connective tissue-related disorders

## Abstract

Fragile X syndrome (FXS) is the most common inherited cause of intellectual disabilities and the second most common cause after Down syndrome. FXS is an X-linked disorder due to a full mutation of the CGG triplet repeat of the *FMR1* gene which codes for a protein that is crucial in synaptogenesis and maintaining functions of extracellular matrix-related proteins, key for the development of normal neuronal and connective tissue including collagen. In addition to neuropsychiatric and behavioral problems, individuals with FXS show physical features suggestive of a connective tissue disorder including loose skin and joint laxity, flat feet, hernias and mitral valve prolapse. Disturbed collagen leads to hypermobility, hyperextensible skin and tissue fragility with musculoskeletal, cardiovascular, immune and other organ involvement as seen in hereditary disorders of connective tissue including Ehlers–Danlos syndrome. Recently, *FMR1* premutation repeat expansion or carrier status has been reported in individuals with connective tissue disorder-related symptoms. We examined a cohort of females with features of a connective tissue disorder presenting for genetic services using next-generation sequencing (NGS) of a connective tissue disorder gene panel consisting of approximately 75 genes. In those females with normal NGS testing for connective tissue disorders, the *FMR1* gene was then analyzed using CGG repeat expansion studies. Three of thirty-nine females were found to have gray zone or intermediate alleles at a 1:13 ratio which was significantly higher (*p* < 0.05) when compared with newborn females representing the general population at a 1:66 ratio. This association of connective tissue involvement in females with intermediate or gray zone alleles reported for the first time will require more studies on how the size variation may impact *FMR1* gene function and protein directly or in relationship with other susceptibility genes involved in connective tissue disorders.

## 1. Introduction

Fragile X syndrome (FXS) is the most common inherited cause of intellectual disabilities and autism spectrum disorder. FXS is the second most common cause of intellectual disability after Down syndrome [1,2]. FXS is an X-linked disorder due to a full mutation of the CGG triplet repeat in the 5′-untranslated region of the fragile X messenger ribonucleoprotein 1 (*FMR1*) gene, formerly named fragile X mental retardation 1, located at Xq27.3, and the most prevalent cause of intellectual disability in males. It affects 1 in 5000 to 7000 men, and 1 in 4000 to 6000 women [1,3,4]. The findings include neuro-behavioral disturbances, communication and social deficits with intellectual disability and facial dysmorphism. Other physical features are suggestive of a connective tissue disorder involving ligaments, muscles, the skeleton and the genitourinary, cardiovascular and immune systems [5,6].

Reported physical findings in FXS include a long, narrow face which is seen in 83% of affected individuals and more commonly in adults. Macrocephaly is seen in 50% to 81% of individuals; prominent ears in 75%; a prominent jaw in 80% seen mostly in adults; flat feet in 29% to 69%; joint hypermobility in 57% and less commonly in adults; palmar and plantar creases, hernias and mitral valve prolapse in >20% of cases; and macro-orchidism in 95% with onset in adolescence or during adulthood (e.g., [1,7,8,9,10,11]). The *FMR1* protein (FMRP) is encoded by the X-linked *FMR1* gene. This protein is crucial in maintaining functions of extracellular matrix-related proteins and key for the development and function of neuronal and connective tissue including collagen. Similarly, disturbed collagen leads to clinical outcomes seen in hereditary disorders of connective tissue (e.g., Ehlers–Danlos syndrome (EDS), Marfan syndrome, osteogenesis imperfecta) (e.g., [12]). These heterogeneous groups of genetic disorders are due to numerous distinct mutations or variants. They are characterized by variable expressivity of joint hypermobility, hyperextensible skin and tissue fragility with musculoskeletal, cardiovascular, immune and other organ system involvement as similarly recognized in those affected with FXS (e.g., [1]).

The normal number of triplet repeats in the *FMR1* gene is less than 45. As this trinucleotide repeat expands beyond this range, the gene is potentially disturbed and may impact the function of the encoded protein. The intermediate or gray zone range consists of 45 to 54 repeats. The premutation or carrier status represents those with 55 to 200 repeats. The premutation can lead to a reduction in FMRP which is crucial in maintaining functions of extracellular matrix-related proteins including matrix metallopeptidase 9 and elastin supporting involvement in the clinical presentation of those females with connective tissue disorders. In addition, elevated *FMR1* mRNA can also cause protein sequestration and result in RNA toxicity which may affect several gene families and proteins involving multiple organ systems along with immune mediation and inflammation seen in those with connective tissue disorders [13,14,15]. Due to expansion to full mutation with over 200 CGG repeats, the *FMR1* gene becomes epigenetically hypermethylated and shuts down, leading to a deficit of its encoded protein (FMRP). An abnormal *FMR1* gene repeat expansion seen in premutation alters protein levels impacting neuronal and connective tissue structure and function (e.g., [6]), leading to hypermobility and other features of a connective tissue disorder (CTD) [5].

To evaluate the potential connection between the triplet repeat expansion in the *FMR1* gene and presentation of connective tissue disorder-related symptoms, we performed CLIA/CAP polymerase chain reaction (PCR) testing of *FMR1* CGG repeats and AGG interruptions—which may induce stability of the CGG expansion region by decreasing further expansion in females with premutation (e.g., [16,17])—in a cohort of females with features of a CTD. These individuals had undergone prior clinical genetic evaluation and testing of a CTD gene-specific panel using next-generation sequencing (NGS), but no potentially damaging variants were identified in genes included on this panel. Herein, we present *FMR1* gene testing results in females with features of a CTD and previous normal DNA testing for connective tissue disorders in the clinical setting and compared to the general population.

## 2. Results

We report our experience utilizing a patient cohort consisting of 100 consecutive unrelated patients presenting for genetic evaluation for a CTD using NGS of a customized CTD gene panel ordered and undertaken at Fulgent Genetics (Temple City, CA) and describe three females with a gray zone *FMR1* gene allele. The most common reason for referral was a suspected connective tissue disorder (n = 71), followed by EDS (n = 18), joint hyperflexibility (n = 7), Marfan syndrome (n = 3) and Chiari malformation (n = 1). Eighty of the one hundred patients were females, and forty-eight were found to not have any pathogenic, likely pathogenic or predicted potentially pathogenic variants using a customized connective tissue disorder gene panel [18]. DNA was then available to test for *FMR1* gene CGG repeat expansions in 39 of the 48 females as features of connective tissue disorders have been reported in those affected with fragile X syndrome (e.g., [5]). The average age (SD) of the females tested for the *FMR1* repeat expansion was 33y (13), with an average (SD) Beighton score of 5.9 (1.5). Sixteen of the females had no variants identified using NGS of hereditary disorders of connective tissue-related genes, while twenty-three females were found with variants of unknown clinical significance but not meeting pathogenicity criteria and interpreted as non-disease-causing.

Three of the thirty-nine females tested for the *FMR1* repeat expansion in our patient cohort were found with a 1:13 ratio of having an intermediate repeat expansion (45–54 repeats) (see Table 1). A chi-square test with Yates’s correction was calculated comparing this intermediate or gray zone frequency to that reported by Tassone et al. [19] in a representative sample involving several ethnic groups in the United States having a 1:66 ratio or 105 of 6889 newborn females studied. The chi-square test value was 6.02 with one degree of freedom, and the one-tailed *p*-value was 0.007; the two-tailed *p*-value was 0.014, with both being significant based on the threshold of *p* < 0.05. The report by Tassone et al. [19] in a large cohort of newborn females was selected as relevant for estimating the frequency of *FMR1* gene findings in the general population.

## 3. Discussion

Several clinical findings and physical features seen in FXS can overlap with those diagnosed with a hereditary connective tissue disorder such as hypermobile EDS. The findings in EDS do include a long, narrow face, ear and jaw problems, flat feet with joint hypermobility, loose skin, hernias and mitral valve prolapse as similarly seen in FXS (e.g., [1,2,7,8,9,10]). EDS has been classified into 13 subtypes and related to 19 separate genes (e.g., [12]) showing all three inheritance patterns (autosomal recessive, autosomal dominant or X-linked). Overall, EDS is considered an uncommon inherited disease with an estimated prevalence ranging from 1 in 2500 to 5000 individuals including the classic form. Other forms of connective tissue disorders are less common (e.g., Marfan syndrome), with a prevalence of 1 in 5000 to 10,000 individuals. Some types of EDS are exceedingly rare such as EDS type VIIC, dermatosparaxis type (OMIM:255410), with fewer than 20 patients reported (e.g., [20]).

Females in our study presented with features of a connective tissue disorder, and potentially related findings including immune/autoimmune problems such as fibromyalgia showed the frequency of an intermediate *FMR1* repeat expansion to be significantly higher than in the general population ascertained through surveying newborn females [19]. For example, 45% of *FMR1* premutation carrier females were found in other studies to have an immune-mediated disorder (IMD) with autoimmune thyroiditis (24%) followed by fibromyalgia (10%) in a survey of 344 carrier females (age 19 to 81y) [15]. Autoimmune thyroid disorder and fibromyalgia have also been reported to be associated with women having a premutation and fragile X-associated tremor/ataxia syndrome (FXTAS) (e.g., [21]). Additionally, about one fifth of those females with premutations had fragile X premature ovarian insufficiency (FXPOI), and one fourth had chronic muscle pain. The pathogenesis of premutation disorders is apparently related to a gain-of-function effect from 1.5- to 8-fold elevated levels due to an expanded CGG repeat *FMR1* mRNA, leading to RNA toxicity [22]. Toxicity is thought to be due to sequestration of RNA-binding proteins, including DROSHA and DGCR8 which are critical for maturing miRNAs including a role in processing *FMR1* [23]. The role of miRNA in regulating autoimmunity in T-cell lymphocytes with potential consequences of miRNA dysregulation in developing IMDs may be due to abnormal control of T-cell function and upregulation of the innate immune response through increased or prolonged inflammatory cytokine production [24]. In addition, *FMR1* premutations lead to dysregulation of the hypothalamic–pituitary axis and enhanced release of the stress hormone cortisol, impacting the immune system and resulting in inflammation including of the central nervous system [25]. Mis-splicing of a variety of messages could lead to several forms of autoimmunity including MS, SLE and RA [26]. Furthermore, miRNA dysregulation was found to be associated with infertility and corpus luteum failure as a possible role in FXPOI secondary to RNA toxicity [15,27].

Previously, Tassanakijpanich et al. [6] also noted an association between CGG repeat expansions in the *FMR1* gene and connective tissue problems in five females with a hypermobile EDS phenotype and having a premutation carrier status (allele sizes in these five females ranged from 66 to 150 CGG repeats) and RNA toxicity compounding problems for connective tissue through mitochondrial dysfunction and inflammation. Hence, our study of females with features of a connective tissue disorder with negative NGS testing of known connective tissue disorder genes and *FMR1* gene studies found a 1:13 ratio for an intermediate allele status.

No premutation cases were found in our study, but an intermediate expansion was seen in a subset of females with connective tissue problems. Fragile X AGG analysis has provided risk predictions for 45–69-repeat alleles having intermediate and small premutation alleles based on fragile X repeat instability upon transmission [28]. The number of AGG interruptions and the length of uninterrupted CGG repeats at the 3′ end have been correlated with repeat instability upon transmission. Maternal alleles with no AGGs conferred the greatest risk for unstable transmission. The magnitude of repeat expansion was larger for alleles lacking AGG interruptions. The number of AGG interruptions in our three females with gray zone alleles each showed two interruptions which is an expected number or range as reported in female premutation cohorts studied (e.g., 57 females with premutations had a mean ± standard deviation = 0.7 ± 0.7, with a range from 0 to 2) [17].

Many genes and their encoded proteins are important for the development of features seen in connective tissue disorders and will require further characterization, with secondary gene effects potentially playing a role. Gene–gene interactions and collagen-related protein or biological pathways that may or may not be disturbed should be studied in those with features of CTD with or without recognized CTD gene involvement. Potential sex effects were raised by Tassanakijpanich et al. [6] due to the observation that most patients presenting with connective tissue disorders are females, showing sex skewness as many more females inherit fragile X premutations compared with men [19]. The role of X chromosome inactivation should also be considered with level of disease involvement for X-linked genes in females. Sex steroid differences do occur between the sexes, with variable effects of sex steroids on muscle tone and tendon/ligament strength being notably different in males and females, further suggesting a significant role in the expression of CTD findings, with more females affected. Furthermore, pain perception and modulation differ between sexes, which may impact musculoskeletal pain which is observed as a feature most often in females affected with a CTD [29].

The status of the gray zone or intermediate repeat expansion and clinical involvement is not clearly established, but evidence suggests an expanding role for gray zone allele involvement with a similar phenotype and manifestations, as discussed and seen in females with the premutation status [30]. For example, the clinical phenotype of adult fragile X gray zone allele carriers in our study is similar to that observed with the *FMR1* gene premutation status including neurological, molecular and cognitive aspects [31]. Gray zone alleles have been reported to expand over two generations to a full mutation, but this typically takes over two generations [32]. Additionally, signs of parkinsonism and earlier death have been reported in adult males having the gray zone allele [33], while studies show that those with premutation alleles are associated with fragile X-associated primary ovarian insufficiency in females, motor ataxia or movement disorders and, more recently, connective tissue-related disorders (e.g., [5,6,34]). We now report a possible association of connective tissue involvement in females with the intermediate or gray zone allele, but additional studies are needed.

The authors would welcome investigations to further understand the role of the *FMR1* gene and altered FMRP with disturbed relationships involving connective tissue development and function. Notably, 60% of the females presenting for genetic services and NGS testing of a CTD-specific gene panel in our report did not have any potentially damaging variants identified in CTD recognized genes. By using leftover DNA available following these initial screens, we were able to readily examine the *FMR1* status using the PCR methodology and follow-up with AGG interruption studies from a commercially based laboratory conducting genetic testing (Fulgent Genetics). Gray zone alleles and FMRP should be studied to determine if altered or reduced protein levels are present which could also impact potential therapeutic interventions. Hence, special emphasis should be placed on the disturbance of the *FMR1* mRNA and encoded protein in those with the intermediate or gray zone allele, if present, and how this size variation may affect *FMR1* gene function directly or its relationship with other inter-related or susceptibility genes including for connective tissue disorders.

## 4. Materials and Methods

### 4.1. Subjects

Over a 3.5-year interval between 2016 and 2020, 100 unrelated consecutive patients were referred for genetic evaluation with features of a connective tissue disorder and seen at the University of Kansas Medical Center (KUMC) Genetics Clinic directed by one of the coauthors (MGB), an ABMG board-certified clinical geneticist. There were 80 females and 20 males, with an average age (±SD) of 33 ± 14 years and an age range of 7 to 68 years [18]. The average Beighton hyperflexibility score was 5.9 ± 1.9 in this cohort, with a number greater than or equal to 5 considered to be abnormal in young adults. Age, gender and medical and family histories were obtained along with recorded physical and phenotypic features. Those identified with associated features of hereditary connective tissue disorders included a positive Beighton hyperflexibilty score (i.e., 5 out of 9 measures) followed by easy bruising, stretchable thin skin with poor scarring, scoliosis and joint dislocations, instability or pain.

### 4.2. Molecular Genetics and Case Reports

#### 4.2.1. Molecular Genetics

Approximately 75 genes are recognized to cause hereditary connective tissue disorders, as examined using a comprehensive connective tissue disorder gene panel with NGS performed at Fulgent Genetics. This is a Clinical Laboratory Improvement Amendments (CLIA) approved and accredited commercial laboratory using established guidelines required for certification following informed consent obtained from all participants prior to collection of buccal samples for DNA extraction. The NGS data included gene name, inheritance, variant type (missense, nonsense, frameshift, indel), coding position, amino acid substitution, zygosity and pathogenicity status (pathogenic, likely pathogenic, unknown clinical significance) based on the American College of Medical Genetics (ACMG) recommendations (http://www.acmg.net/ (8 February 2022)). For each gene variant, the likelihood of being protein-damaging and influencing the expression of a disorder-related phenotype (potentially pathogenic) was determined and calculated using approximately ten in silico prediction programs, amino acid evolutionary conservation reported in primate and mammal genome databases, variant and allele frequencies found in human genomic databases (e.g., Broad Institute GnomAD) and Grantham distance scores greater than 100 for missense variants following standard protocols (e.g., [18]). Variants not meeting internal quality control standards were confirmed by Sanger sequencing.

*FMR1* gene CGG triplet repeat sizes were determined using a CLIA/CAP-approved PCR methodology on the same DNA sample performed by Fulgent Genetics previously with NGS of CTD gene-specific panels. Those with *FMR1* gene repeat expansion had AGG interruptions analyzed using the FMR1 Asuragen kit (Austin, TX). There were 48 females out of 80 tested, and no ACMG-classified pathogenic, likely pathogenic or predicted potentially pathogenic variants were identified in genes from the comprehensive connective tissue disorder panel. Of these 48 females, 39 had sufficient DNA remaining for *FMR1* gene repeat expansion testing. The full *FMR1* repeat expansion is seen in 99% of patients with FXS [35]. Of the 39 females, 3 showed a gray zone *FMR1* gene variant status, while the other 36 showed a normal *FMR1* gene pattern. A description of the clinical, medical, genetic and family history information of the three females can be found below, and the laboratory flow chart for the genetic testing results is illustrated in Figure 1.

#### 4.2.2. Case Reports

##### Case #1

A 26-year-old female was seen in the Genetics Clinic at KUMC for evaluation of a connective tissue disorder. She presented with hypermobility, polyarthralgia, easy bruisability and joint laxity of both major and minor joints. She complained about joint pain, a tingling sensation and hand and lower back pain with extensive hyperflexibility, particularly of her wrists, shoulders, elbows and knees. She had right shoulder subluxation which required repair on two separate occasions along with repair of labral tears of both hips. She had loose ligaments/capsules repaired on her left shoulder. She has persistent tachycardia and was diagnosed with hypothyroidism. She had a heart murmur in the past, but cardiac evaluation found no structural defects or anomalies. She has a history of weekly migraine headaches.

A three-generation pedigree was obtained. Her parents are both alive but adopted with no available family histories. She has an older brother with psychiatric and behavioral problems. No consanguinity was noted.

On physical examination, she was well developed and nourished with a normal blood pressure and heart rate. Her weight was 55.3 kg (40th percentile) and height was 160 cm (25th percentile), with a body mass index of 21.6 kg/m^2^. Her head circumference was 54 cm (40th percentile), total hand length was 17.7 cm (50th percentile) and middle finger length was 7.2 cm (40th percentile). Her Beighton hyperflexibility score was 7 out of 9, indicating a possible connective tissue disorder with a score greater than 5. She exhibited positive signs for hyperextensible bilateral knees, bilateral fifth digits and bilateral hyperextensible thumbs to wrists and placed both palms on the floor upon standing and bending forward. The rest of the physical examination was normal including abdomen, HEENT (head, eyes, ears, nose and throat), neurological, pulmonary, cardiac, cutaneous and psychiatric.

Clinical genetic testing was ordered using next-generation sequencing (NGS) for CTDs based on her physical examination and medical history (as described above). These test results as a component of their medical care showed no pathogenic, likely pathogenic or potentially pathogenic variants and were interpreted as normal. This test was then followed by an *FMR1* gene triplet repeat expansion study using an approved PCR of available DNA, and a normal allele (20) and an intermediate (51) repeat expansion size allele were found. AAG interruptions were also studied, and one AGG interruption was found prior to 20 repeats which could belong to either allele. A second AGG interruption occurred after 20 repeats, which would belong to the 51-repeat allele.

##### Case #2

A 45-year-old female was seen in the Genetics Clinic at KUMC for evaluation of a connective tissue disorder. She presented with migraine headaches, carpel tunnel syndrome and dislocated thumbs requiring previous surgical intervention during adulthood, decreased muscle mass, non-trauma-related ankle and heel sprains, spontaneous nose bleeds and easy bruising, thin loose skin with poor scarring, myopia and astigmatism. She also presented with bipolar disorder, depression, insomnia, gastro-esophageal reflux disease (GERD), asthma and vitamin D deficiency.

She was born without any congenital anomalies but exhibited hypermobility since childhood. She has a history of heart palpitations and syncope. She has a history of hidradenitis involving several subcutaneous sweat glands that were removed from the axillary regions.

A three-generation pedigree was obtained and showed that she has two sons (17 and 12 years of age), with her older son having bipolar disorder and joint laxity. She has a 36-year-old brother with a history of depression. Her mother died at 70 years of age due to leukemia. Her mother also had a history of nose bleeds, joint laxity, high blood pressure and severe depression. Her maternal grandmother died at the age of 50 years due to heart issues, and her maternal grandfather died at the age of 70 years due to a stroke. Her father is 74 years of age and has high blood pressure, a pacemaker and dementia. Her paternal grandmother died from a stroke and dementia. Her paternal grandfather died from an unexplained heart disease. No consanguinity was noted.

On physical examination, her blood pressure and heart rate were within normal range. Her weight was 84.1 kg (80th percentile) and height was 174 cm (90th percentile), with a body mass index of 27.8 kg/m^2^. In addition, her head circumference was 57 cm (60th percentile), ear length was 6.1 cm (55th percentile), total hand length was 18.8 cm (75th percentile) and middle finger length was 7.7 cm (75th percentile). Her Beighton heteroflexibility score was 3 out of 8 (had left thumb surgery and could not score for mobility). Her right shoulder was lower than her left shoulder upon standing. The rest of her physical examination was within normal range other than the features noted historically above. The *FMR1* gene triplet repeat expansion study using PCR of available DNA showed a normal allele (33) and an intermediate (48) repeat expansion size allele. AAG interruptions were also studied, and one AGG interruption was found prior to 33 repeats, which could belong to either allele. A second AGG interruption occurred after 33 repeats and would belong to the 48-repeat allele.

##### Case #3

A 44-year-old female was seen in the Genetics Clinic at KUMC. She presented with concerns about fibromyalgia, tachycardia, migraines, tinnitus and hypertension with subluxation of her major and minor joints caused by a connective tissue disorder such as EDS.

A three-generation pedigree was obtained and showed that both of her sons (13 and 16 years of age) had features of a connective tissue disorder including postural orthostatic tachycardia (POTs). Her 16-year-old nephew has scoliosis and elbow dislocations. Her 72-year-old mother has a history of Churg–Strauss vasculitis, and her 73-year-old father has a history of six separate abdominal hernias. Her maternal grandmother died in her 80 s from an unknown cause. Her maternal grandfather passed away at a young age with a history of a heart attack and high cholesterol. Her paternal grandmother and paternal grandfather both died at the age of 70 years with a history of breast cancer and mesothelioma, respectively.

On physical examination, her blood pressure was 114/72 and heart rate was 99. Her weight was 91.4 kg (95th percentile) and height was 166 cm (70th percentile), with a body mass index of 33.1 kg/m^2^. Her Beighton hyperflexibility score was 4 out of 6. She has a history of loose, thin and stretchable skin and numerous vertical striae on her abdomen. The rest of her physical examination was within normal range including HEENT, pulmonary/cardiac, neurological and psychiatric. The *FMR1* gene triplet repeat expansion study using PCR of available DNA showed a normal allele (24) and an intermediate (50) repeat expansion size allele. AAG interruptions were also studied, and two AGG interruptions were found, with both occurring after 24 repeats and therefore belonging to the 50-repeat allele.

### 4.3. Statistical Analyses

A chi-square test with Yates’s correction was used for statistical analysis of the *FMR1* triplet repeat status in this study to determine differences in the frequency of females in our cohort with no significant connective tissue gene variants identified and their *FMR1* status. We then performed comparisons with the reported *FMR1* gene repeat status in the general population of females. The threshold for statistical significance was set at *p* < 0.05.

## Figures and Tables

**Figure 1 ijms-23-09090-f001:**
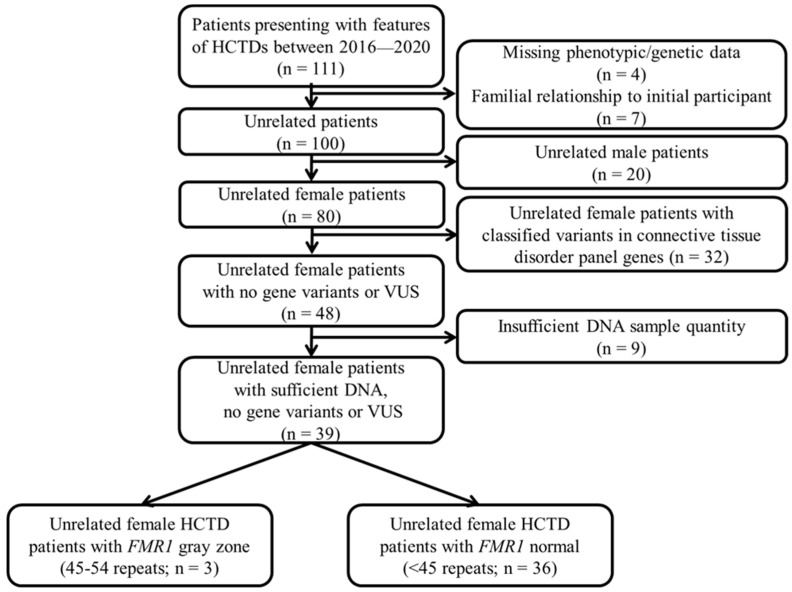
*FMR1* gene repeat testing in female patients presenting for heritable connective tissue disorders (HCTDs). Flow chart with details for patients who presented to the genetics clinic over a 3.5-year period. Next-generation sequencing results for genes included on commercially available connective tissue disorder testing panels were evaluated for 80 unrelated female patients. Unrelated female patients with no variants or variants of unknown significance (VUS) reported as ACMG-classified pathogenic, likely pathogenic or determined potentially pathogenic based on allele frequency, biological conservation, Grantham distance and damaging in silico predictions were followed up with *FMR1* triplet repeat testing using an approved polymerase chain reaction (PCR), given a sufficient DNA sample was available.

**Table 1 ijms-23-09090-t001:** Molecular and clinical findings in three adult females with FMR1 gray zone repeats in a cohort of females presenting for genetic services.

Participants	Age (Years)	CGG Repeats	AGG Interruptions (Number; Allele Size)	Immune-Mediated Disorder or Inflammation	Beighton Hyperflexibility Score
Case #1	26	20, 51 *	(1; 20 or 51 *) (1; 51 *)	Polyarthralgia, hypothyroidism, migraines	7 out of 9
Case #2	45	33, 48 *	(1; 33 or 48 *) (1; 48 *)	Positive anti-nuclear antibody (ANA), migraines	3 out of 8
Case #3	44	24, 50 *	(0; 24) (2; 50 *)	Fibromyalgia, polymyositis, lupus, migraines, family history of Churg–Strauss vasculitis	4 out of 6

* Designates the intermediate or gray zone repeats.

## Data Availability

Data sharing is not applicable to this article as the original data presented in this study are included, further inquiries can be directed to the corresponding authors.

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
