# Peer review of "Connective Tissue Disorders and Fragile X Molecular Status in Females: A Case Series and Review"

_ijms, 2022, doi:10.3390/ijms23169090_

Round 1

Reviewer 1 Report

Fragile X syndrome (FXS) is caused by a deficiency of the FMR1 gene “Fragile X messenger ribonucleoprotein 1” gene’s encoded protein Fragile X Messenger Ribonucleoprotein", or FMRP, formerly, the fragile X mental retardation protein (FMRP). The protein  is  crucial  in maintaining functions of extracellular matrix related proteins, key for development and function of connective tissue including collagen. 

Individuals with FXS show physical features suggestive of a connective  tissue disorder including loose skin and joint laxity, flat feet, hernias and mitral valve  prolapse. Disturbed collagen leads to hypermobility, hyperextensible skin and tissue fragility with musculoskeletal,  cardiovascular  and  other  organ  involvement  as  seen  in  hereditary  disorders  of connective tissue including Ehlers-Danlos syndrome. Recently, FMR1 mutation repeat expansion or carrier status has been reported in individuals with presentation of connective tissue disorder-related symptoms. 

This study examined a cohort of females with features of a connective tissue disorder presenting for genetic services using next generation sequencing (NGS) of a connective tissue disorder gene panel consisting of approximately 75 genes. In those females with normal NGS testing for connective tissue disorders, the FMR1 gene was then analyzed using repeat expansion studies. Three (age range 26-49) of  39  females  or  1:13  ratio  was  found  to  have  gray  zone  or  intermediate  CGG alleles (51, 48, 50, respectively) that  were significantly  different  (p<0.05)  when  compared  with  females  from  the  general  population with a 1:66  ratio.  The authors concludes that this association  of  connective  tissue  involvement  in  females  with  the  intermediate  or  gray zone alleles  for the first time will require more studies on how the size variation may impact  FMR1 gene function directly or in relationship with other susceptibility genes including for connective tissue disorders.

This is an important effort by the authors to study the intermediate expansion of the FX gene category that may also be associated with connective tissue problems as suggested in other studies with the premutation status. The authors are commended to zero in on the FX gene’s proposed link here. The case report study needs work under a major revisions as follows. 

1) The authors should used the FX-gene new nomenclature…Fragile X syndrome (FXS) is caused by a deficiency of the FMR1 gene “Fragile X messenger ribonucleoprotein 1” gene’s encoded protein Fragile X Messenger Ribonucleoprotein", or FMRP.

2) In this study under the review, the authors say “that noted an association between full mutation CGG-repeat expansions in the FMR1 gene and connective tissue  problems  and  found  five  females  with  hypermobile  EDS  phenotype  having  a premutation carrier status (55-200 CGG expansion). “

The statement of “an association between full mutation CGG-repeat expansions in the FMR1 and connective tissue  problems”  the authors then end their sentence  with “…and  found  five  females  with  hypermobile  EDS  phenotype  having  a premutation carrier status (55-200 CGG expansion).“ 

So, how is that, is this a mosaic, meaning that some cells have the FMR1 full mutation and some the FMR1 premutation? Is that what the authors meant here? 

Otherwise it looks confusing, and incorrect to state as is written currently. 

It has to be clarified since FXS means that there are over 200 CGGs *full mutation), and a epigenetic mechanism kicks in leads to the FMRP deficiency, which underlies the impaired function of connective tissue including collagen.      

3) The authors rightfully cite a recent study by Tassanakijpanich et al. [6] but they fail to use its value to a full extent needed here (see below).

4) To expand on the aforementioned, Tassanakijpanich et al. Those five cases were female and ranged between 16 and 49 years. The range of CGG-repeat allele sizes ranged from 66 to 150 repeats. These authors also write that “The premutation can lead to a reduction of FMRP, which is crucial in maintaining functions of the extracellular matrix-related proteins, particularly matrix metallopeptidase and elastin.”

This study under the review the authors conclude “Hence, our study of females with features of a connective tissue disorder with negative NGS testing of known connective tissue disorder genes and FMR1 gene expansion found a 1:13 ratio of intermediate status.”  

But this paper under the review discussion needs to be expanded as follows.

Tassanakijpanich et al. also point toward the FMRP underlying the connective tissue issue. Data have shown that in the FX premutation, the FMRP can be affected even with CGGs 120, especially 150 and over. Indeed, in Tassanakijpanich et al., 3/5 subjects had CGGs over 100 (105, 119, and 100-150), which can explains the FMPR-connective issue link. 

These authors actually suggested four possible mechanisms may lead to connective tissue problems for fXPCs: (1) FMRP deficiency, (2) mRNA toxicity and (3) secondary gene effects, and 4) Potential sex effect, or perhaps all three suggested mechanisms are compounded.

For this study under the review, 2, 3, and 4 should be extensively discussed and cited accordingly. 

5) Suggest that authors generate a table that would summarize key molecular findings form those 3 case studies they describe. Currently, it is not easy to follow, and take time reader to find the info, while this is only a case report of 3 subjects.

Again, Tassanakijpanich et al. is a good model to follow, including to generate a modified version of their Table 1.  

6) Suggest to contrast this study with your study in more details, it is worthwhile for an interested reader. Including to contrast the FX genetic test this study used “FMR1 gene triplet repeat analysis was undertaken on the same DNA sample by Fulgent Genetics utilizing approved methodology.” 

What does that mean as for the FX test, and how is that similar and different from  Tassanakijpanich et al? They used a combination of PCR and Southern blot analysis to determine the CGG sizing and methylation pattern on isolated genomic DNA. 

Reviewer 2 Report

Dear Authors,

your paper discuss the association between intermediate FMR1 alleles in females with connective tissue disorders, describing the cases of three females from independent families.

Major comments.

1. If you assert (lines 115-117) that intermediate FMR1 alleles are significantly higher in females with connective tissue disorders than in the general population, you have to perform the same analyses on 39-40 females from general population. In fact, if we consider immune/autoimmune problems this statement is not correct.

2. FMRP is a crucial protein in synaptogenesis and in maintaining a specialized form of neuronal ECM such as perineural nets. It is an RNA-binding protein that interacts with so many RNAs and other proteins, also in the cellular ECM (connective tissue). In lines 14-15 (Abstract) the role of FMRP should be appropriately rephrased.

3. The phrase in lines 66-68 is quite cryptic and contradictory with respect to what was previously stated: in which way FMR1 repeat expansion could impact connective structure and function if before you mentioned the role of FMRP as ECM interactor? You may hypothesized that reduced FMRP levels (that could be present in premutation and intermediate alleles) may interfere with connective function and structure, although FMRP levels are not quantified in the three patients.

4. The "approved methodology" to assess FMR1 sizing (section 4.2.1) should be properly described. Is this PCR method able to detect AGG interruptions? How many AGG interruptions within the intermediate allele are present (if any) in the three females? You could add a comment in the Discussion.

Minor comments.

- some typos should be corrected (i.e. number "4disabilites" in line 37; the phrase "e.g." before the numbers' refs sthe square brackets hould be deleted within the text);

-in the Introduction the word "disability" seems more appropriate than "disabilities" (from line 37 to 43);

-the three case reports (paragraph 4.2.2) could benefit from a shift in the Results section (paragraph 2);

-the phrase "More studies are needed" in line 125 should be removed, because it is repeated soon after in line 138.

Round 2

Reviewer 1 Report

Thanking the authors for their effort to address this reviewer's suggestions.  

Few more compelling minor suggestions are advised to be addressed. 

Lines 16, 17 “In addition to neurological and behavioral problems, individuals with FXS…”

 In addition to neurobehavioral (an alternative is to say neuropsychiatric) and behavioral problems…” 

(the point is that except seizure in FXS, which is an example of a pure neurological, the vast majority of presenting in FXS is neurobehavioral, neuropsychiatric). 

Lines 37-38 Should read Fragile X syndrome (FXS)…..and autism spectrum disorder. 

Line 53, The FMR1 protein… FMR1 should be italic (represents the gene) whereas the FMRP abbreviation is not italic.

Indeed, line 212, the authors use the term “altered FMRP-protein..”

In addition, her, line 212, they actually may just say “altered FMRP..” as it has the term protein it its abbreviation. 

Line 74

..” than 200 and the gene becomes fully disturbed with no FMRP produced..”

Its kind of an awkward statement…anyway, suggest to say…due to the expansion full-mutation, the FX gene gets epigenetically (hypermethylation) shut off, which leads to a deficit of its encoded protein: FMRP.

Lines 74-76 “An abnormal FMR1 gene repeat expansion seen in premutation, and possibly intermediate alleles  could alter  protein  levels  impacting  neuronal  and..

At this point what we know, is that CGG premutation repeats over 120, clearly more 150 and above, alter the FMRP levels. Thus, it is not true to say “possibly intermediate alleles  could alter the protein  levels” as stated in the Introduction.

Perhaps in the Discussion section to say “possibly intermediate alleles  could alter the protein  levels” but it is rather speculative at this point.    

Like on lines 218-220, this is a fair statement to use in Future directions 

“Gray  zone  alleles  and  FMRP should be  studied to  determine  if  altered  or  reduced  protein  levels  are  present which could  impact potential  therapeutic  interventions.” The statement also confirms the issue raised for the statement on the lines 74-76 above.

.

Author Response

Reviewer 1.

 Thank you for your kind consideration, All changes requested by reviewer 1 are incorporated in the manuscript

Reviewer 2 Report

Dear Authors,

your revised version of the paper entitled "Connective tissue disorders and fragile X molecular status in females: a case series and review" fits with the suggested revisions. The paper may now be considered for publication in IJMS. 

Author Response

Reviewer 2. 

Thank you.

 Did not have any suggested changes